

# DaGzang: a synthetic data generator for cross-domain recommendation services

Luong Vuong Nguyen[1], Nam D. Vo[1] and Jason J. Jung[2]

[1] Department of Artificial Intelligence, FPT University, Da Nang, Vietnam
[2] Department of Computer Engineering, Chung-Ang University, Seoul, Korea

## ABSTRACT

Research on cross-domain recommendation systems (CDRS) has shown efficiency by leveraging the overlapping associations between domains in order to generate more encompassing user models and better recommendations. Nonetheless, if there is no dataset belonging to a specific domain, it is a challenge to generate recommendations in CDRS. In addition, finding these overlapping associations in the real world is generally tricky, and it makes its application to actual services hard. Considering these issues, this study aims to present a synthetic data generation platform (called DaGzang) for cross-domain recommendation systems. The DaGzang platform works according to the complete loop, and it consists of the following three steps: (i) detecting the overlap association (data distribution pattern) between the real-world datasets, (ii) generating synthetic datasets based on these overlap associations, and (iii) evaluating the quality of the generated synthetic datasets. The real-world datasets in our experiments were collected from Amazon's e-commercial website. To validate the usefulness of the synthetic datasets generated from DaGzang, we embed these datasets into our cross-domain recommender system, called DakGalBi. We then evaluate the recommendations generated from DakGalBi with collaborative filtering (CF) algorithms, user-based CF, and item-based CF. Mean absolute error (MAE) and root mean square error (RMSE) metrics are measured to evaluate the performance of collaborative filtering (CF) CDRS. In particular, the highest performance of the three recommendation methods is user-based CF when using 10 synthetic datasets generated from DaGzang (0.437 at MAE and 0.465 at RMSE).

## INTRODUCTION

Recommendation services aim to model user preferences based on user history interactions, such as item ratings (*Nguyen et al., 2020a*; *Hong & Jung, 2018*; *Nguyen, Jung & Hwang, 2020b*; *Vuong Nguyen et al., 2021*). Some of the traditional methods in recommendation systems (RSs) such as matrix factorization (MF) (*Koren, RM & Volinsky, 2009*; *Salakhutdinov & Mnih, 2007*) or neural collaborative factoring (CF) (*Cheng et al., 2016*; *Dziugaite & Roy, 2015*; *He et al., 2017*; *Nguyen, Nguyen & Jung, 2020c*) are applied to several specific datasets collected from several sources. The recommended results of these methods achieve acceptable accuracy. The most important goal in an RS is to increase the accuracy of items recommended to users (*Vuong Nguyen et al., 2021*). This means the RS

Corresponding author
Jason J. Jung, j2jung@gmail.com

must be designed to collect as much information as possible from users. However, data is increasing daily in the real world through social media and e-commercial sites. This leads to the data having a lot of noise, useless information, *etc.* That is challenging for recommendation services with two major problems, cold-start issues, and data sparsity. Many recent studies have proposed solutions for these problems, but they are still not completely solved. One of the practical solutions showing efficiency in overcoming the sparsity and cold-start problems is deploying cross-domain recommendation services (CDRS) by using overlapped associations between different domains. In particular, the CDRS learns the latent features between users and items from multiple domains. These latent features enhance the recommendation in the target domain (*Taneja & Arora, 2018*). Therefore, CDRSs generally generate recommendation items with improved accuracy even with a sparse dataset of a target domain (*Vo, Hong & Jung, 2020*).

The rapid development of methods for CDRSs has facilitated enriched datasets from different domains to evaluate the performance of CDRSs. Unfortunately, the data are not always available, especially from the specific domain for testing the recommender techniques. Table 1 shows a statistical of several real-world datasets. The density information in this table shows that these datasets has sparsity problem. Besides, comparing the number of user-item and the number of ratings in each dataset, we can see the lack of ratings in these datasets. We propose a novel method for generating synthetic datasets to overcome this problem. This method follows two scenarios and is described as follows.

- The data distribution is pre-defined such that a system manager or end-user customizes a new dataset by choosing any data attribute in terms of the statistical distribution or data sparsity.
- Domain-specific features are applied to the domain adaptation technique (*Bousmalis et al., 2016*) to extract the rating patterns from a real-world dataset in a specific domain, and the patterns are then used to generate synthetic data with a certain number of shared users/items from the input data.

According to these scenarios above, we deploy the synthetic dataset generator, called DaGzang (http://recsys.cau.ac.kr:8084/dakgalbi). The DaGzang platform allows the end users to customize several parameters at the input stage of the data-generating procedure. Furthermore, DaGzang has domain adaptation inside, which means that if the end users customize the parameters, the whole platform automatically re-calculates synthetic datasets adapted to the two scenarios mentioned. Moreover, DaGzang is the platform of the DakGalBi framework, which refers to the CDRS presented in our previous study (*Vo & Jung, 2018*; *Vo, Hong & Jung, 2020*). DakGalBi is a comprehensive framework, from input to output, which consists of functions to overcome the traditional cold-start and sparse issues in recommendation services. In conclusion, the DaGzang platform generates various datasets with multiple domain features, and the conventional CF algorithms deployed in the DakGalBi subsequently process the datasets to produce the recommendation items. Notably, with the DaGzang generator platform, researchers can leverage different datasets to test the recommendation system's performance or evaluate its recommendation method.

The main contributions of this study can be summarized as follows.

**Table 1  Statistical analysis of the existing datasets.**

| Dataset | #Users | #items | #Ratings | Density |
|---|---|---|---|---|
| Amazon Automotive | 850,418 | 319,112 | 1,373,768 | 0.0005% |
| Amazon Baby | 530,890 | 63,426 | 915,446 | 0.0027% |
| Book-Crossing (*Ziegler et al., 2005*) | 104,283 | 339,556 | 1,149,780 | 0.0032% |
| Amazon Digital Music | 477,235 | 265,414 | 836,006 | 0.0007% |
| Amazon Toys and Games | 1,341,911 | 326,698 | 2,252,771 | 0.0005% |
| Amazon Instant Video | 425,922 | 22,965 | 583,933 | 0.0061% |
| Amazon Videos Games | 825,767 | 49,210 | 1,324,753 | 0.0033% |
| MovieLens | 282,228 | 52,889 | 27,753,444 | 0.1859% |
| Serendipity 2018 (*Nguyen et al., 2018*) | 103,661 | 48,151 | 9,997,850 | 0.2003% |

- The DaGzang platform is proposed to generate synthetic datasets that have several functions consisting of creating multiple models for users, providing various statistical distributions, randomizing sets of items/ratings, and preserving domain-specific features.
- The synthetic datasets generated by DaGzang are deployed to the CF CDRS (in the DakGalBi framework) to obtain the recommendations provided to evaluate the CDRS performance. These datasets are also published online and allow other researchers to evaluate their recommendation method for the CDRS in the future.

The rest of the paper is structured as follows. 'Related Work' presents the background of the synthetic data and literature reviews associated with this concept. The synthetic data generator DaGzang is described in 'DaGzang: a Synthetic Data Generator' including the methods used in various scenarios to produce the synthetic data. The experimental results, evaluation, and discussion are provided in 'Experimentation'. Finally, in 'Conclusions', we conclude the current research results and show the direction of future work.

## RELATED WORK

Data regarding human activities and behaviors are time-consuming and difficult to compile. In addition, their availability is often limited. To model everyday behaviors and physiological parameters, previous attempts to construct simulations of human activity data based on sensors focused on mathematical models such as Markov chains and Petri networks (*Virone et al., 2003*; *Hoag & Thompson, 2007*). Later works combined these methods with other modeling methods to simulate much more complex data. For example, *Helal, Mendez-Vazquez & Hossain (2009)* applied Markov chains to construct the generated timestamps to model activity patterns combined with a Poisson distribution. Later, this study became a fully realized framework for building and sharing synthetic datasets with scientific groups (*Helal et al., 2011*). Other recent works employed generative adversarial networks (GANs) to produce synthetic data. This method uses two differentiable functions represented by neural networks. The generator network creates data from some probability distributions, and the discriminator network decides if the input comes from the generator or the actual training set. The generator maximizes the probability of making the

**Table 2  Comparison between the MovieLens 20M dataset and a large-scale industrial dataset.**

|               | MovieLens 20M | Large-scale industrial dataset |
| ------------- | ------------- | ------------------------------ |
| #users        | 138,000       | Hundreds of millions           |
| #items        | 27,000        | 2,000,000                      |
| #topics       | 19            | 600,000                        |
| #observations | 20,000,000    | Hundreds of millions           |

discriminator mistake its inputs as accurate, while the discriminator guides the generator to produce more realistic images (*Goodfellow et al., 2014*).

Machine learning algorithms and techniques require larger sets of labeled data for realistic testing. However, the supply of actual reality-labeled data is limited. Therefore synthetic data should be generated to implement machine learning more efficiently. For example, by using GANs to extend the training data size and diversity for liver lesion classification, *Frid-Adar et al. (2018)* successfully improved the classification efficiency. To maximize the precision of one nearest neighbor (1-NN) dynamic time warping classifier, *Forestier et al. (2017)* used a weighted variant of the time-series averaging process to enlarge training time-series datasets. Other research that has been investigated for synthetic data generation poses problems when generating random variables of independently and identically distributed data. The data shows high dimensionality and high complexity levels. Furthermore, recent dataset implementations in academia and at the industrial production systems scale reveal vast differences in dataset sizes. Table 2 shows more details about this issue, which was inspired by the comparison between the MovieLens 20M dataset (*Harper & Konstan, 2016*) and an industrial dataset (*Zhao et al., 2018*). The number of users, items, and topics in the MovieLens dataset are much smaller than that in the industrial dataset (*Zhao et al., 2018*).

Researchers leveraged the advantages of synthetic data to bridge this gap. Notably, in terms of using synthetic datasets for recommendation systems (RS), *Belletti et al. (2019)* proposed a new approach by expanding pre-existing public datasets. The properties of the dataset, including the distribution of user engagements, the popularity of items, and the item/user interaction matrices singular value spectra, are preserved when generating recommendation system data *via* fractal expansion using Kronecker Graph Theory (*Leskovec et al., 2005*; *Leskovec et al., 2010*). However, this study aimed to develop large synthetic datasets for a specific domain, such as the MovieLens 20M. In addition, with the new solution for the RS, there was some research such as CD-SPM, and CDRec-CAS (*Anwar & Uma, 2019*; *Anwar, Uma & Srivastava, 2023*). These studies focus to solve problems in cross-domain RS and this lead to the need for synthetic cross-domain datasets to verify the performance of the proposed methods more effectively.

Inspired by these related works, in this paper, we introduce a synthetic data generator called DaGzang to create synthetic datasets for different specific domains. In our experiments, combined with synthetic data generation, we deployed these synthetic datasets to the DakGalBi (http://recsys.cau.ac.kr:8084/dakgalbi) framework to verify their

suitability for evaluating the cross-domain recommender systems based on traditional CF algorithms.

# DAGZANG: A SYNTHETIC DATA GENERATOR

This section provides detailed information on the DaGzang platform, including the description, architecture, and functionalities.

## DaGzang architecture

DaGzang was implemented in the Java programming language that contains the web service in the front end and background service in the back end. In the background service, we deployed the MySQL and Java server pages (JSP) with the Spring Framework. Then, we applied the model-view-controller (MVC) model to handle the large datasets generated by multiple thread processing. In addition, our system has a simple process and exemplary performance in extensive data analysis geared toward adaptivity with many datasets and users. The Spring Web MVC framework is designed around a DispatcherServlet that handles all the HTTP requests and responses. The request processing workflow of the Spring Web MVC DispatcherServlet is illustrated in the following diagram.

As shown in Fig. 1, the sequence of events corresponding to an incoming HTTP request to the *DispatcherServlet* is described as follows: (1) The *DispatcherServlet* consults the *HandlerMapping* for calling the *Controller* after receiving an HTTP request. (2) The *Controller* receives the request and calls the service methods (GET or POST) that set model data based on defined business logic and returns the view name to the DispatcherServlet. (3) The *DispatcherServlet* interacts with the *ViewResolver* to select the defined view for the request. (4) The *DispatcherServlet* proceeds to transfer the model data to the view that is finally shown on the browser.

In designing the front end of DaGzang, the interaction with users is carefully considered to construct the simplest process for users to generate suitable datasets. Specifically, our system is web-based, and reducing latency in every process is the most important goal. Furthermore, to prevent the loading of all datasets in the system, DaGzang applies the pagination method in the displayed results and follows three rules in designing the user interface (*Mandel, 1997*), which are (i) placing users in control, (ii) decreasing user memory loading, and (iii) making a consistent interface. Following these rules, we use one template for all pages to guarantee the consistency of the user interface. Therefore, regardless of their actions, people only remember the same signs (*e.g.*, fonts, keys, and items).

Consequently, user comfort with the system is paramount. The interface is also the same for all actions (*e.g.*, using the same approach to view results). The system also clearly describes the processes for generating the data, allowing users to easily recognize where step they are staying, what they are processing, and what they need to do next. In addition, DaGzang provides users with roll-back procedures to return to past steps if they have incorrect actions.

Figure 2 shows the interface of the data generation function in our DaGzang system. Based on each purpose, the users can decide which parameters are injected from the list in the configure functions on the left. This list of parameters includes the number of users,

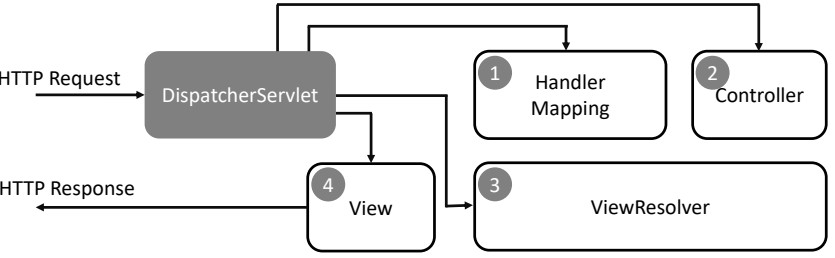

**Figure 1   Request processing workflow of the Spring Web MVC DispatcherServlet.**

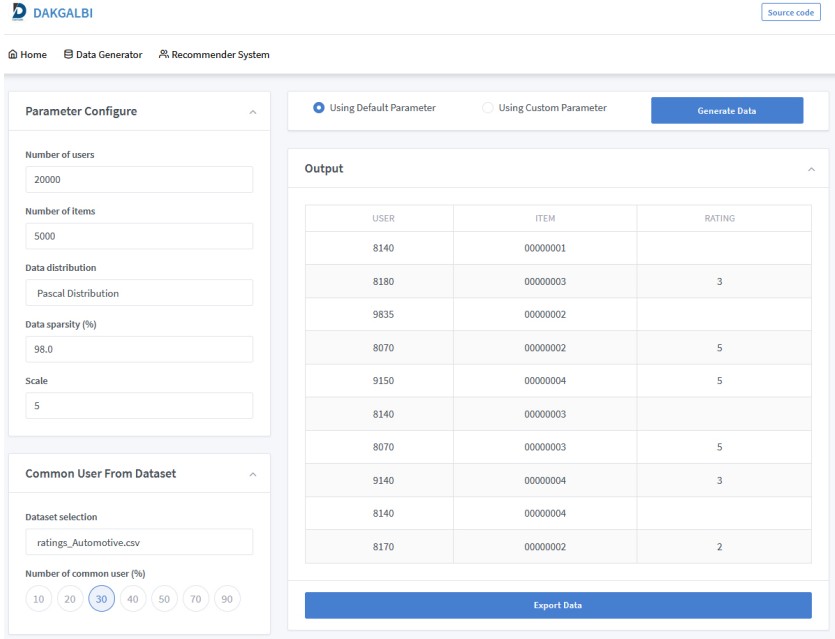

**Figure 2   Screenshot of the data generator interface of the DaGzang platform.**

items, and similar users as well as the sparsity, distribution, scale, and selection of the data. All these parameters were depicted as follows.

- Number of users or items: number of users or items in the output dataset.
- Data sparsity: how many rating values are empty in the output dataset.
- Scale: the range value of ratings.
- Data distribution: this parameter decides which distribution the output dataset has.
- Dataset selection: this parameter is used to select a real-world dataset to extract the rating patterns that can transfer to the output dataset. This function is used when the user wants to generate datasets based on a specific domain feature.
- Number of common users: This parameter indicates how many common users are between a real-world dataset and the output dataset. Particularly, the system uses

this parameter to calculate the number of common users in the synthetic dataset corresponding with the rating patterns extracted from the input dataset.

Then, when the user clicks on the generate data button, our data generation algorithms will perform the calculations and display the synthetic data results in the output screen on the bottom right.

## DaGzang functionalities

The key function of DaGzang is to generate synthetic datasets according to the users' configurations and adapt to their demands. Creating synthetic datasets involves three main steps: (i) extracting data patterns (*i.e.,* statistical distributions, sparsity, and rating patterns) from the selected real-world datasets, (ii) injecting user-customized parameters, and (iii) executing the generator based on these customized parameters combined with the extracted data patterns.

To generate a realistic synthetic dataset, we integrate all the steps described above and construct the algorithm. Firstly, the real-world dataset was extracted pattern (rating patterns, statistical distributions, sparsity) and set as initial input. This step aims to create the initial dataset and increases them depending on user purposes according to the pattern extracted from the real-world dataset. In the next step, the ratings for all users are randomly generated for this dataset in order to construct the user preferences. Then, in the last step, the domain adaptation was used to map the preferences between common/similar users. In this step, the rating was updated again based on the injecting user-customized parameters combined with the extracted patterns (ratings) in the first step. In particular, given a real-world dataset, by $\mathbf{A} = \{D, S, R_p\}$, we indicate the set of all attributes, where $D$ and $S$ represent the distribution and sparsity, respectively; $R_p$ denotes the rating pattern of the real-world dataset. Let $I_l$ with $l \in (1, L)$ be the set of all possible *items* in the $l$-th domain. For each *user*, there is a maximum of $n$ ratings for $n$ *items* in domain $I_l$. Then, it is possible to denote a set of ratings in $I_l$ of the form $R_l = \{r_{l_1}, r_{l_2}, \ldots, r_{l_n}\}$, where $r_{l_n}$ is the rating of the user for the $n$-th item in the domain $I_l$. Therefore, for each *user*, we have two sets of the *item* and *rating* represented by $I_l$ and $R_l$, respectively, such that $item\_rating = \{I_l, R_l\}$. Algorithm 1 presents the procedures to generate synthetic data that is described as follows

1. Initial extraction of the attributes from a real-world datasets to get $\mathbf{A}$.
2. Generate randomly for a set of $item\_rating = \{I_l, R_l\}$ for all users.
3. Define number of ratings will be created in the synthetic dataset based on distribution $D$ and sparsity $S$.
4. Generate dataset based on parameters from step 2 and 3.
5. Update ratings for $sh_u$ users.

---
**Algorithm 1: Synthetic data generation**

---
**Input**: A real-world sample dataset: **IN** dataset
**Output: SYNTHETIC dataset**
Step 1: Initial data pattern from a real-world dataset
**initialization**:
    $A = \{D, S, R_p\} \Longleftarrow$ IN dataset
    $sh_u \Longleftarrow$ **user input**
Step 2: Generate ratings for all users randomly
**for** each *user* **do** {**making set** $\{I_l, R_l\}$ }
    **Calculate the number of users**: from **S**, calculate the number of users $m$
    **repeat**
        Set user $k = 1, k \, \epsilon \, [1, m]$;
        Randomize all elements in $\{I_l, R_l\}$;
        Update the user distribution: $= D$;
        $k = k + 1$;
    **until** $k \, = m$
**end for**
Step 3: Use domain adaptation to map preferences of common users
**for** each *user* in $sh_u$ **do**
    **Mapping rating patterns**:
    **repeat**
        set user k $= 1, k \, \epsilon sh_u$
        from extracted rating patterns, update the ratings for user *k;*
        $k \, = k + 1;$
    **until** $k \, = sh_u$
**end for**

In order to allow users to customize the output dataset, the system provides several options consisting of the data distribution based on the user-defined, domain-specific feature data and real-world input data models. *The user-defined data distribution* allows the end-user to select the desired data distribution to generate, while *domain-specific feature data* allows the user to choose the domain-specific data that the system is expected to generate based on the specific domain.

*The real-world input data model* option directs the system to extract the data model from real-world input data to create the synthetic data generation rule. Adapting to these requirements, DaGzang is designed with two separate modules. The first is the module to generate synthetic datasets with user input parameters. The second module is a pool that stores many existing well-known datasets, allowing users to choose between using an existing dataset and creating a new one for themselves.

## EXPERIMENTATION

This section shows the results of our experiments and evaluations and provides a discussion of the results to demonstrate the usefulness of the DaGzang platform. Our experimental purposes are described as follows.

---

 

1. Generate synthetic datasets by DaGzang based on extracting overlapping associations between real-world datasets.
2. Deploy these synthetic datasets in the DakGalBi CDRS to show the usefulness of completely synthetic datasets for recommendation algorithms.
3. Construct numerical results by using a bipartite graph and binary search to show the amount of overlap between real-world datasets and the dataset generated by DaGzang.
4. Comparing the proposed method with CART (*Drechsler & Reiter, 2011*)

## Experimental setup

To conduct the experiments, we first collect 13 real-world datasets from Amazon (https://jmcauley.ucsd.edu/data/amazon/) and insert them into the DaGzang platform. Then, we classify these datasets into different types of the domain according to the information relevant to these datasets. The detailed information of these datasets is shown in Table 3

DaGzang extracts the overlap association between these real-world datasets in the next step. This overlap association is defined as the rating pattern of users and is labeled in our database. We then generate the synthetic dataset by combining this overlap with some parameter configurations in the input of DaGzang, and the platform loops this step according to the parameter changes to obtain a list of synthetic datasets. For this experiment, we generate 90 synthetic datasets with several parameter configurations, as illustrated in Table 4. Finally, each synthetic dataset generated from DaGzang is used for the recommendation functions in DakGalBi (the CF CDRS we built). The recommendation methods used for this experiment are the CF, including the item-based CF, user-based CF, and user-item (matrix factorization) CF. The mean absolute error (MAE) and root mean squared error (RMSE) metrics were used to calculate the accuracy of these output recommendations. The formulations of MAE and RMSE are described as follows:

$$\mathbf{MAE} = \frac{1}{n}\sum_{i=1}^{n}\left|y_i - y_i^p\right| \text{ and } \mathbf{RMSE} = \sqrt{\frac{1}{n}\sum_{i=1}^{n}\left(y_i - y_i^p\right)^2} \tag{1}$$

where $n$ is the number of items, $y_i$ is the real ratings and $y_i^p$ is the predicted ratings. The results of MAE and RMSE range from 0 to infinity. Infinity is the maximum error according to the scale of the measured values.

We first use 10 of 90 datasets generated from DaGzang, including multiple domains for each experiment, to calculate the MAE and RMSE. The number of datasets increases each time, running ten until the entire dataset consists of 90. The experimental results are used to evaluate the accuracy of recommendation methods in DakGalBi CDRS. We then discuss the efficiency and suitability of these generated synthetic datasets with the CF CDRS to clarify whether DaGzang adapts to our purpose. To accomplish this, we use the bipartite graph and binary search to find the best matching. We observed that the number of common users in the rating patterns represented the number of common users in the synthetic and real datasets. In other words, these values show how much overlap exists between these datasets. Besides, we compare the proposed method with another, CART (*Drechsler & Reiter, 2011*). For this experiment, we use the 10 synthetic datasets generated from each method. Then we

**Table 3 Real-world datasets in DaGzang.**

| # | Dataset Name | Domain | Range | #ratings |
|---|---|---|---|---|
| 1 | Health-and-Personal-Care | Health | 1–5 | 2,982,326 |
| 2 | Grocery-and-Gourmet-Food | Grocery, Foods | 1–5 | 1,297,156 |
| 3 | Digital-Music | Music | 1–5 | 836,006 |
| 4 | Clothing-Shoes-and-Jewelry | Fashion | 1–5 | 5,748,920 |
| 5 | Cell-Phones-and-Accessories | Electronics | 1–5 | 3,447,249 |
| 6 | CDs-and-Vinyl | Music | 1–5 | 3,749,004 |
| 7 | Automotive | Automotive | 1–5 | 1,373,768 |
| 8 | Amazon-Instant-Video | Movies | 1–5 | 583,933 |
| 9 | Baby | Fashion | 1–5 | 915,446 |
| 10 | Musical-Instruments | Music | 1–5 | 500,176 |
| 11 | Patio-Lawn-and-Garden | Home | 1–5 | 993,490 |
| 12 | Toys-and-Games | Toys, Games | 1–5 | 2,252,771 |
| 13 | Video-Games | Games | 1–5 | 1,324,753 |

**Table 4 Parameter configurations.**

| # | Parameter | Value |
|---|---|---|
| 1 | Sparsity | 99% |
| 2 | Rating scale | 5 |
| 3 | Number of users | 10K, 20K, 30K, 40K, 50K |
| 4 | Number of items | 20K, 30K |
| 5 | Number of statistical distributions | Normal, Poison, Pascal |
| 6 | Number of ratings | 20K, 50K, 100K, 200K, 500K |
| 7 | Number of common users | 10%, 20%, 30% |

use these two datasets to input the DakGalBi cross-domain recommendation system with CF-based recommendation methods. Finally, we calculated the MAE and RMSE each time running the recommendations from DakGalBi. All experimental results and discussions are presented in the next section.

## Experimental results and discussion

After generating 90 datasets following the randomization of the changing parameters shown in Table 4, we separate these datasets into nine parts consisting of [1-10], [1-20], [1-30], [1-40], [1-50], [1-60], [1-70], [1-80], and [1-90]. Nine groups of datasets are provided for the second experiment embedded in the DakGalBi CDRS to build the recommendations based on CF algorithms. The results of the second experiment are shown in Table 5. As shown in the experimental results, as the number of synthetic datasets increases, the overlap association between domains decreases and the accuracy of the recommendation methods is also reduced. Following each metric, MAE and RMSE, we can see in Figs. 3 and 4, the

**Table 5** The results of MAE and RMSE in terms of the accuracy of recommendation methods in the CDRS using synthetic datasets generated from DaGzang.

| # syntheticdatasets | User-based CF | | Item-based CF | | SVD | |
|---|---|---|---|---|---|---|
| | MAE | RMSE | MAE | RMSE | MAE | RMSE |
| 10 | 0.437 | 0.465 | 0.449 | 0.466 | 0.503 | 0.511 |
| 20 | 0.466 | 0.502 | 0.478 | 0.503 | 0.522 | 0.532 |
| 30 | 0.501 | 0.554 | 0.513 | 0.542 | 0.546 | 0.601 |
| 40 | 0.525 | 0.601 | 0.518 | 0.544 | 0.613 | 0.633 |
| 50 | 0.581 | 0.643 | 0.541 | 0.560 | 0.664 | 0.682 |
| 60 | 0.642 | 0.663 | 0.596 | 0.704 | 0.708 | 0.715 |
| 70 | 0.763 | 0.702 | 0.657 | 0.733 | 0.775 | 0.796 |
| 80 | 0.765 | 0.767 | 0.775 | 0.776 | 0.822 | 0.830 |
| 90 | 0.841 | 0.833 | 0.798 | 0.801 | 0.874 | 0.881 |

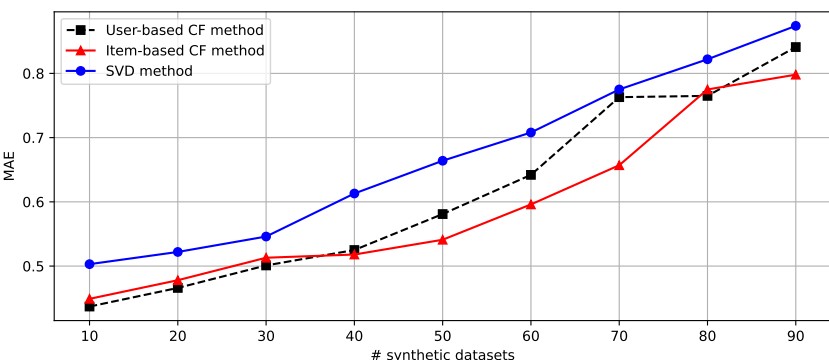

**Figure 3** The comparison between Item-based, User-based, and SVD methods in the CDRS in terms of MAE metric.

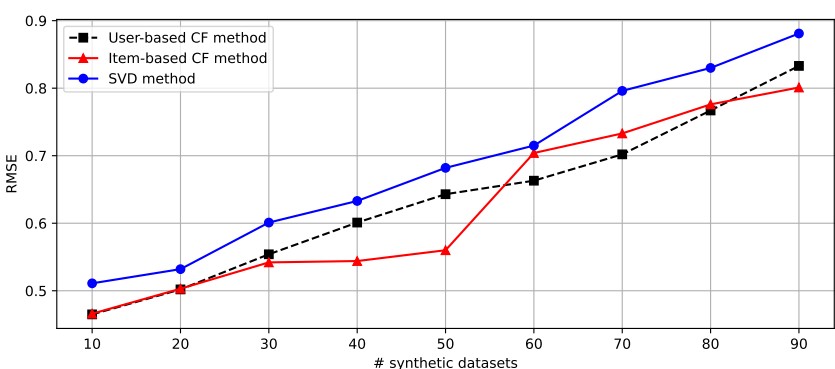

**Figure 4** The comparison between Item-based, User-based, and SVD methods in the CDRS in terms of RMSE metric.

**Table 6  The results of calculating the number of common users between synthetic datasets.**

| # datasets | 10 | 20 | 30 | 40 | 50 | 60 | 70 | 80 | 90 |
|---|---|---|---|---|---|---|---|---|---|
| # common users | 787 | 685 | 607 | 511 | 429 | 333 | 245 | 153 | 60 |

**Table 7  The comparison results between DaGzang and CART synthetic data generation methods of the accuracy of recommendation methods in the CDRS in terms of MAE and RMSE.**

| | MAE | | RMSE | |
|---|---|---|---|---|
| | DaGzang | CART | DaGzang | CART |
| User-based CF | 0.437 | 0.747 | 0.465 | 0.788 |
| Item-based CF | 0.449 | 0.755 | 0.466 | 0.791 |
| SVD | 0.503 | 0.789 | 0.511 | 0.793 |

errors increase depending on the number of synthetic data used. In most cases from 10 to 90 synthetic datasets, these results are linear.

However, considering the case of ten synthetic datasets, the performance of the recommendation methods is very impressive, even with the traditional user-based CF method. This demonstrates that the proposed method to generate a synthetic dataset in DaGzang is suitable for evaluating the recommendation algorithms. Furthermore, we deploy the third experiment using the bipartite graph and binary search to clarify the overlap between these synthetic datasets. The numerical results of this experiment are shown in Table 6. In addition, for the last experiment that compared the proposed method with CART, the proposed method showed outperformance. This experimental result is described in Table 7

The experimental results show that the generated synthetic datasets adapt to the evaluation purpose. The datasets generated from DaGzang can be used to evaluate CDRS algorithms, although the accuracy is not sufficiently high. However, the main purpose of this study is to overcome the scarcity of collected real datasets for the CDRS. DaGzang provides functions to handle requirements from users based on several scenario purposes, such as (i) generating datasets for single-domain, (ii) multi-domain, and (iii) cross-domain implementations. To accomplish this, in DaGzang, we allow users to configure the parameters according to their purposes, highlighting our platform's critical performance.

## CONCLUSIONS

In this paper, we proposed a new approach to generate synthetic datasets by extracting specific domain features from common users. First, we used existing datasets to derive data attributions such as distribution, sparsity, and rating patterns. We then leveraged the domain adaptation technique to transfer users' rating patterns from existing real-world datasets to synthetic datasets. With the help of this technique, the real-world input dataset is utilized to share content with the generated synthetic dataset in the output. Moreover, the randomization function creates attributes for the synthetic dataset in terms of either the data distribution or sparsity. Finally, we designed the DaGzang synthetic data generator, an online web application, to deploy the proposed data-generating method. By implementing

efficiency techniques, the system can generate synthetic datasets from given specific existing datasets with parameters injected by users. In addition, the system can evaluate the quality of synthetic datasets by applying efficient cross-domain recommendation algorithms. We also proposed a method to detect common user patterns in two datasets by finding the best matchings using bipartite graph theory and binary search. The experiments not only demonstrated the effectiveness of the DaGzang platform in generating data with various output requirements and diverse input parameters and computing MAE and RMSE with high accuracy but also revealed the significance of the proposed method in detecting the associations between datasets. For future work, we aim to extend the DaGzang and validated the performance of the synthetic data generated from DaGzang with other synthetic data simulators.

### Funding

This study was supported by a grant from the "Leaders in INdusty-university Cooperation 3.0" Project by the Ministry of Education and National Research Foundation of Korea. The funders had no role in study design, data collection and analysis, decision to publish, or preparation of the manuscript.

### Grant Disclosures

The following grant information was disclosed by the authors:
Ministry of Education and National Research Foundation of Korea.

### Competing Interests

The authors declare there are no competing interests.

### Author Contributions

- Luong Vuong Nguyen conceived and designed the experiments, performed the experiments, analyzed the data, performed the computation work, prepared figures and/or tables, authored or reviewed drafts of the article, and approved the final draft.
- Nam D. Vo conceived and designed the experiments, performed the experiments, analyzed the data, performed the computation work, prepared figures and/or tables, authored or reviewed drafts of the article, and approved the final draft.
- Jason J. Jung conceived and designed the experiments, performed the experiments, analyzed the data, authored or reviewed drafts of the article, and approved the final draft.

### Data Availability

The source code is available at GitHub and Zenodo: https://github.com/kecau/DaGzang.
Luong Vuong Nguyen, & Jason J. Jung. (2023). DaGzang: A Synthetic Data Generator for Cross-domain Recommendation Services (1.0). Zenodo. https://doi.org/10.5281/zenodo.7800378.

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
