# Peer review of "DaGzang: a synthetic data generator for cross-domain recommendation services"

_PeerJ Computer Science, doi:10.7717/peerj-cs.1360_

## Round 0.1 · original submission · Major Revisions

Dear Authors,

Please revise and resubmit your manuscript. Also, please improve the English language presentation. Thank you.

·

Basic reporting

-If possible, the Literature review is summarized with a table.
-A lot of bad English constructions, grammar mistakes, and misuse of articles: a professional language editing service is strongly recommended (e.g., the ones recommended by IEEE, Elsevier, and Springer) to sufficiently improve the paper's presentation quality for meeting Peerj's high standards.
-It is suggested: - Increase the relevant and recent cited works like (CD-SPM, MRec-CRM, CDRec-CAS, etc.).
- Abstract and conclusion require significant readability and improvement to make them more forceful.

Experimental design

What types of overlap did you use user overlap, item overlap, and user-item overlap?

Validity of the findings

overall fine

Additional comments

NA

Reviewer 2 ·

Basic reporting

The manuscript is focused on synthetic data generator for recommendation system. The authors tried to evaluate the performance of the data generator on number of datasets using Collaborative Filtering techniques which is novel. But there are few concerns about the manuscript as listed below.

1. Abstract can be written in a more informative way to clearly specify the purpose or need of such data generator. Also if the outcome of the experiment is stated in the abstract, it will make it meaningful.
2. Provide the evidence for the claim made on line 59 "However, increasing the data information leads to two major problems in traditional RSs, cold- start issues and data sparsity, that are challenging for recommendation services"
3. Explanation of the term "Density" is missing in the manuscript but it is used in table1.
4. Sentence construction and grammar need to be checked. e.g. Line 71-72 "By comparing the number of user items and the ratings, we can see the lack of ratings in these datasets", Line 108-110 "Next, the synthetic data generator, DaGzang, is described in Section 3, including the methods used in various scenarios to produce the synthetic data." These sentences can be written using professional English for international audience.
5. On line 133-135, the reference cited (Goodfellow et al., 2014) is used for the images which is irrelevant for the work done in the experiment . Instead, it would be beneficial to include relevant references.
6. Table 2 is written as Table 1. Parameter Configurations. Please check the same.
7. Line 173- Java Programming Language looks appropriate than Java Program Language
8. Line 202-204, "The system also clearly states the procedure to generate the data, so users can quickly identify where they stay, what they process, and what they need to do next". Here, please check the sentence construction.
9. Line 197- "reducing the memory loading of users" can be written in a better way.

Experimental design

1. The explanation of Algorithm1 looks very superficial. It would be better if it is explained in detail for every step.
2. Also explanation of the parameters used for the mentioned distributions is unclear which can be added to enhance the quality of the manuscript.

Validity of the findings

1. The results are validated using User and Item based CF and SVD for the generated datasets using MAE and RMSE. In order to prove the effectiveness of the DaGzang, it would be good if the same results are obtained using other synthetic data simulator like SynRec and then compared with the DaGzang's performance.
2. A graphical representation of the result would be more appropriate along with results in table.

Additional comments

Overall, It would be better to proof read the manuscript once for the grammar and sentence construction.

·

Basic reporting

The author proposed method for A Synthetic Data Generator for Cross domain Recommendation Service (DaGzang:) mainly used DakGalBi with collaborative filtering (CF) algorithms, user-based CF, and item-based CF. The experimental analysis is not well explained, and the proposed method is worthy for investigation. The paper is lack of small issue which should be consider for improving the manuscript.


1. The Author did not clarify any particular problem statement in this manuscript.
2. The Abstract part need to some modification (Author did not mention any Mathematical data regarding that)
3. The motivation of the paper is not clear it should be mentioned clearly on abstract.
4. The author must focus on the small typos, uses of punctuation and English level throughout the manuscript.
5. I will suggest author to use more exiting method for comparison of the proposed method to show the result.
6. Need to modify references also, most of the references are old.

Experimental design

no comment

Validity of the findings

no comment

Additional comments

The author proposed method for A Synthetic Data Generator for Cross domain Recommendation Service (DaGzang:) mainly used DakGalBi with collaborative filtering (CF) algorithms, user-based CF, and item-based CF. The experimental analysis is not well explained, and the proposed method is worthy for investigation. The paper is lack of small issue which should be consider for improving the manuscript.


1. The Author did not clarify any particular problem statement in this manuscript.
2. The Abstract part need to some modification (Author did not mention any Mathematical data regarding that)
3. The motivation of the paper is not clear it should be mentioned clearly on abstract.
4. The author must focus on the small typos, uses of punctuation and English level throughout the manuscript.
5. I will suggest author to use more exiting method for comparison of the proposed method to show the result.
6. Need to modify references also, most of the references are old.

---

## Round 0.2 · Minor Revisions

Please revise and resubmit the manuscript with the changes requested by the reviewer.

·

Basic reporting

The authors addressed all the comments properly.

Experimental design

Ok

Validity of the findings

Everything fine

·

Basic reporting

The author proposed method for A Synthetic Data Generator for Cross domain Recommendation Service (DaGzang:) mainly used DakGalBi with collaborative filtering (CF) algorithms, user-based CF, and item-based CF. The experimental analysis is updated according to previous comment, and the proposed method is also highlighted for investigation. The paper is few small issues is pending which should be consider for improving the manuscript.

Experimental design

NA

Validity of the findings

I suggested author to use more existing method for comparison of the proposed method to show the result, it is pending till now (already Author added 379-380 small paragraph regarding this, but it is not enough I think, Author should be compared at least single existing method).

---

## Round 0.3 · accepted · Accept

Authors have addressed all of the reviewers' comments.